# L-MBOP-E: Latent-Model Based Offline Planning with Extrinsic Policy Guided Exploration

## Abstract

Offline planning has recently emerged as a promising reinforcement learning (RL) paradigm. In particular, model-based offline planning learns an approximate dynamics model from the offline dataset, and then uses it for rollout-aided decision-time planning. Nevertheless, existing model-based offline planning algorithms could be overly conservative and suffer from compounding modeling errors. To tackle these challenges, we propose L-MBOP-E (Latent-Model Based Offline Planning with Extrinsic policy guided exploration) that is built on two key ideas: 1) low-dimensional latent model learning to reduce the effects of compounding errors when learning a dynamics model with limited offline data, and 2) a Thompson Sampling based exploration strategy with an extrinsic policy to guide planning beyond the behavior policy and hence get the best out of these two policies, where the extrinsic policy can be a meta-learned policy or a policy learned from another similar RL task. Extensive experimental results demonstrate that L-MBOP-E significantly outperforms the state-of-the-art model-based offline planning algorithms on the MuJoCo D4RL and Deepmind Control tasks, yielding more than 200% gains in some cases. Furthermore, reduced model uncertainty and superior performance on new tasks with zero-shot adaptation indicates that L-MBOP-E provides a more flexible and light-weight solution to offline planning.

## 1 Introduction

Offline planning is an emerging reinforcement learning (RL) approach designed to synergize the strength and flexibility of online planning with the advantages of offline learning. A general consensus is that although online planning algorithms Nagabandi et al. (2020); Chua et al. (2018); Lowrey et al. (2018); Wang & Ba (2019) have demonstrated strong performance, their need for continuous interactions with the environment can incur prohibitively high cost. In contrast, offline RL algorithms Kidambi et al. (2020); Yu et al. (2020); Kumar et al. (2020); Fujimoto et al. (2018) only require access to an offline dataset collected by some behavior policy. Notably, a number of model-free offline RL algorithms have recently been proposed, including SAC-N, EDAC An et al. (2021), PBRL Bai et al. (2022), RORL Yang et al. (2022), and TD3-BC-N Fujimoto & Gu (2021). It is worth pointing out that while these algorithms have demonstrated strong performance, they are not readily amenable to decision-time planning Hamrick et al. (2021). Model-free online planning algorithms, such as MCTSnet Guez et al. (2018), TreeQN Farquhar et al. (2018), and Value Prediction Network Oh et al. (2017), employ a neural network architecture which mirrors a search tree to enable implicit planning. By taking advantage of both online planning and offline learning, model-based offline planning learns an approximate dynamics model from the offline dataset, which can be used to devise planning algorithms for effective system control.

There are a number of challenges in model-based offline planning, including 1) compounding modeling errors due to the inaccuracy and uncertainty associated with the approximate dynamics model learned from limited data, and 2) the distributional shift between the distribution of visited state-action pairs during planning and that of the offline dataset. To tackle these challenges, existing approaches Argenson & Dulac-Arnold (2020); Zhan et al. (2021); Diehl et al. (2021) for offline planning typically employ model-predictive control (MPC) Richalet et al. (1978) to re-plan at each iteration, and use a behavior cloned policy to constrain trajectory rollouts while planning. For exam-

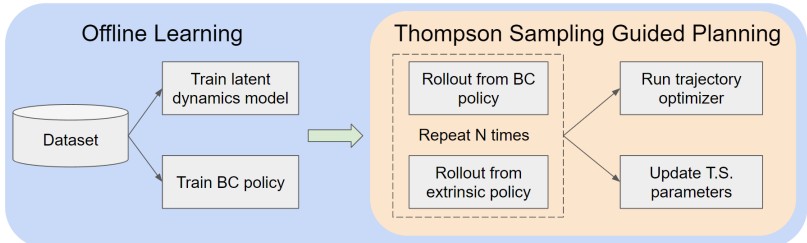

Figure 1: Overview of L-MBOP-E. A latent dynamics model and the BC policy are trained from the offline dataset. Leveraging the latent model, a Thompson Sampling (T.S.) exploration procedure is used to guide exploration following the better one between the BC policy and extrinsic policy.

ple, MBOP Argenson & Dulac-Arnold (2020) and MOPP Zhan et al. (2021) encourage exploration by allowing sampling from the behavior policy with added deviation. Despite the control flexibility, these offline planning algorithms still face several key limitations. (1) *Uncertainty and inaccuracy of high-dimensional dynamics model.* The dynamics models are often trained in the high-dimensional observation space. While re-planning at each state in MPC can help reduce the effect of compounding modeling errors, offline planning still suffers from out-of-distribution errors, because the states visited during planning might differ from the states present in the offline dataset. (2) *Overly conservative planning.* The constrained trajectory rollouts will lead to planned actions that closely follow the behavior policy, which is overly conservative and hinges heavily upon the quality of the BC policy.

To address these two limitations, in this work we propose L-MBOP-E, a model-based offline planning algorithm that makes use of 1) low-dimensional latent dynamics model learning and 2) guided exploration with an extrinsic policy. As shown in Fig. 1, the underlying rationale is as follows: (1) *To deal with the inaccuracy of the dynamics model due to the limited samples in the offline dataset, we advocate a low-dimensional latent representation for the state space, which can yield a higher-accuracy dynamics model and in turn improve prediction quality and hence reduce the likelihood of compounding errors.* This resonates with human's interactions with an environment where we typically do not try to reason directly with the observation space, but rather with some abstracted features of the observations. (2) Further, to mitigate the overly-conservative planning constrained by the BC policy, it is plausible to take advantage of an extrinsic policy which can be another policy obtained from either meta learning or a related task Finn et al. (2017); Lin et al. (2022); Yue et al. (2023); Li et al. (2020). *Based on the online returns of both extrinsic policy and behavior policy, a Thompson Sampling based exploration strategy is proposed to ensure that the planning would mostly follow the guidance of the better policy for specific state-action pairs.* In particular, when both the behavior policy and extrinsic policy are non-expert, there will be regions of the state-space where the behavior policy performs better and other regions where the extrinsic policy performs better. As such, we can expect that the extrinsic policy will complement the behavior policy in some subtle way, and the planning using both policies will enable the algorithm to selectively learn from both policies, leading to improved performance.

## 2 RELATED WORK

Model-based offline planning Argenson & Dulac-Arnold (2020); Zhan et al. (2021) seeks to directly plan actions for execution in an environment by leveraging a learned dynamics model from an offline dataset. This differs from the online setting where one can collect additional data during interaction to improve the dynamics model. MBOP Argenson & Dulac-Arnold (2020) extends the ideas of Planning with Deep Dynamics Models (PDDM) Nagabandi et al. (2020) to the offline setting. In addition to learning an approximate dynamics model, MBOP also trains a behavior cloned (BC) policy from the offline dataset. MBOP uses the BC policy to constrain the explored trajectories in order to reduce distributional shift. However, constraining the explored trajectories to only follow the BC policy limits the potential of planning, therefore MOPP Zhan et al. (2021) attempts to improve exploration by boosting the variance of the actions produced by the BC policy, and using a pruning scheme based on dynamics uncertainty to avoid potential out-of-distribution (OOD) samples. While this approach enhances exploration compared to the base BC policy, its performance still heavily

depends upon the quality of the BC policy. To overcome this challenge, L-MBOP-E uses an extrinsic policy to help guide exploration, in addition to the BC policy.

A number of recent studies on model-based RL have investigated the use of latent state representations with dynamics models, which can capture higher-level features of the environment and facilitate the learning. In related work, TD-MPC Hansen et al. (2022) learns a latent state representation which encodes the dynamics and reward signal, which it then leverages for online planning. Dreamerv3 Hafner et al. (2023) utilizes world models Ha & Schmidhuber (2018) for training a policy with synthetic data. World models encode the history of observations encountered thus far into a hidden state via recurrent neural networks, which is then used for prediction with a latent dynamics model. Due to the additional complexity, world models often require large amounts of data to train. *Different from the above approaches, L-MBOP-E employs a state decoder, in addition to reward signals, to facilitate zero-shot task adaptation, and as a result, the dynamics model employed is lightweight and can be trained offline with minimal data.*

## 3 PROBLEM FORMULATION

### 3.1 MARKOV DECISION PROCESS

As is standard, we consider a Markov Decision Process (MDP) defined by a tuple $(\mathcal{S}, \mathcal{A}, P, r, \gamma)$, where $\mathcal{S}$ is the state space, $\mathcal{A}$ is the action space, $P(s_{t+1}|s_t, a_t)$ is the transition dynamics, $r(s_t, a_t)$ is the reward function, and $\gamma \in (0, 1]$ is the discounting factor. RL aims to learn an optimal policy $\pi^*$ which can maximize the cumulative reward, i.e., $R = \sum_{t=0}^{\infty} \gamma^t r(s_t, \pi^*(s_t))$. Following recent studies Argenson & Dulac-Arnold (2020); Zhan et al. (2021), we fix $\gamma = 1$ and thus only consider the finite-time horizon return. In the offline setting, the algorithm only has access to a static dataset $\mathcal{D}$ of trajectories of the form $\{(s_t, a_t, r_t, s_{t+1})\}$ generated by some behavior policy $\pi_b$.

### 3.2 OFFLINE PLANNING WITH LEARNED DYNAMICS MODELS

**Model-based Offline Planning.** Model-based offline planning methods Argenson & Dulac-Arnold (2020); Zhan et al. (2021) generally learn an approximated dynamics model $f_m$ through supervised learning and then employ a planning algorithm to determine a trajectory with a high return based on this learned model; which is subsequently implemented in the online environment. Meanwhile, a value function $V_b$ is used to extend the planning horizon beyond $H$ steps. Hence, offline planning aims to find an optimal policy $\pi_{op}^*$ such that the accumulated reward is maximized, i.e.,

$$\pi_{op}^*(s_0) = \arg\max_{a_{0:H-1}} \mathbb{E}\left[\sum_{t=0}^{H-1} r(s_t, a_t) + V_b(s_H)\right].$$

Specifically, MBOP Argenson & Dulac-Arnold (2020) learns a behavior cloned policy $f_b(a_{t-1}, s_t)$ as a prior for action sampling when the planning algorithm is rolling out trajectories over the learned model $f_m(s_t, a_t)$.

**Trajectory Optimization.** After having a set of trajectories and the associated return, MBOP employs an extended version of the MPPI Williams et al. (2017) optimizer to obtain the optimal action sequence. Specifically, let $\mathbf{A}_b$ be the set of action trajectories which are sampled using the learned behavior policy $f_b$ and $\mathbf{R}_b = \{R_1, \cdots, R_{|\mathbf{A}_b|}\}$ be the associated cumulative returns. Then the optimized trajectory of actions is obtained by re-weighting the actions in each trajectory according their exponentiated return, i.e.,

$$\mathbf{T}^* = \frac{\sum_{n=1}^{|\mathbf{A}_b|} \exp\left(\kappa \mathbf{R}_b[n]\right)\mathbf{A}_b[n]}{\sum_{n=1}^{|\mathbf{A}_b|} \exp\left(\kappa \mathbf{R}_b[n]\right)},$$

where $\kappa$ is the re-weighting factor.

**Limitations of Existing Approaches.** The performance of existing offline planning algorithms Argenson & Dulac-Arnold (2020); Zhan et al. (2021) is often limited by two factors: (1) the compounding model errors and (2) the over-restrictive planning with the behavior cloned policy. Specifically, in real-world applications such as video games and robotics, the high-dimensional state observations may contain redundant information and thus impose great model uncertainty for RL agents trained from limited offline data. Meanwhile, MBOP samples actions exclusively from the learned behavior policy to relieve the out-of-distribution error during offline learning. However, this over-restrictive planning unavoidably hinders the full utilization of trajectory optimizers such as MPPI, which re-

quires sufficient state-action space coverage in order to perform well Zhan et al. (2021). These issues are still present in MOPP Zhan et al. (2021) even though large deviation is allowed from the behavior policy in sampling.

# 4 LATENT-MODEL BASED OFFLINE PLANNING WITH EXTRINSIC POLICY GUIDED EXPLORATION (L-MBOP-E)

To address the limitations mentioned above, in this section we introduce L-MBOP-E which is built on two key ideas. First, we use a low-dimensional latent state space when training the dynamics model from limited offline data, aiming to mitigate the effects of compounding errors. Second, we propose a Thompson Sampling based exploration strategy with an extrinsic policy $\pi_c$ to guide planning beyond the BC policy, where the extrinsic policy can either be a meta-learned policy or a policy acquired from a similar RL task.

## 4.1 LATENT MODEL REPRESENTATION

Instead of directly learning the dynamics models using the offline dataset, we utilize insights from representation learning literature Srinivas et al. (2020); Chandak et al. (2019); Yarats et al. (2021); Edwards et al. (2018); Watter et al. (2015) and employ latent dynamics models to reduce model uncertainty. The rationale behind incorporating latent models is to reduce dimensionality; allowing for more accurate predictions by capturing the core reasoning in higher-level input domains when using limited samples. Specifically, L-MBOP-E first jointly learns the latent dynamics model and a representation mapping between original and latent state spaces with an encoder-decoder architecture Hinton & Salakhutdinov (2006). Then, a behavior policy and a Q function are learned with the use of the latent state representation. In this regard, the planning algorithm of L-MBOP-E incorporates five parameterized function approximators, i.e.,

- $z_t = e(s_t)$ is the state encoder which maps the observations to the latent space, where we normalize the latent states to lie on the hypersphere to improve convergence.
- $d(z_t) = s_t$ is the state decoder, which decodes the latent state back into the original space.
- $f_m(z_t, a_t) = (z_{t+1}, r_t)$ is the latent dynamics model, which takes as input the current state encoding and an action, and produces the latent representation for the next state and the predicted reward. We use $f_m(z_t, a_t)_z$ and $f_m(z_t, a_t)_r$ to denote the predicted latent state and reward, respectively.
- $f_b(a_t|z_t) = \mathcal{N}(\mu(z_t), \Sigma(z_t))$ is the behavior cloned policy, which is modeled as a Gaussian distribution over the actions for that iteration.
- $Q_b(z_t, a_t)$ is the learned Q function for the underlying true behavior policy $\pi_b$.

**Latent Dynamics Model.** Our proposed method trains the autoencoder and dynamics model jointly. In this way, the network can align the learned latent state representation with the underlying dynamics captured by the latent model. Specifically, denote the joint parameter for the latent model and encoder-decoder to be $\theta$. We design the loss function as follows:

$$\mathcal{L}(\theta|\mathcal{D}) = \sum_{i=1}^{N} \|e(s_{i+1}) - f_m(e(s_i), a_i)_z\|^2 + \lambda_1(r_i - f_m(e(s_i), a_i)_r)^2 + \lambda_2\|s_i - d(e(s_i))\|^2,$$

where the first term trains the dynamics model to predict the latent representation of the next state, the second term trains the dynamics model to predict the instantaneous reward for the given state-action pair, and the last term is the reconstruction loss for the encoder-decoder pair. $\lambda_1$ and $\lambda_2$ are used to balance the importance of each term.

**Behavior Policy and Value Function Learning in the Latent Space.** After training the state encoder and latent dynamics model, the BC policy is trained via maximum likelihood on $\mathcal{D}$ with the latent state representations. The value function $V_b$ is obtained by learning a Q function via Fitted Q Evaluation Le et al. (2019) on the latent representations of $\mathcal{D}$. Let $y_i = r_i + Q_b^{k-1}(z_{i+1}, a_{i+1}), (z_i, a_i, r_i, z_{i+1}, a_{i+1}) \sim \mathcal{D}$. Then at the $k$-th iteration, the Q function is updated as follows,

$$Q_b^k(z_i, a_i) = \min_{f \in \mathcal{F}} \frac{1}{N} \sum_{i=1}^{N} [f(z_i, a_i) - y_i]^2,$$

where $\mathcal{F}$ is the function class and $N$ is the number of samples in the offline dataset $\mathcal{D}$. The value function can be further evaluated by $V_b(s_t) = \mathbb{E}_{a \sim \pi_b}[Q_b(z_t, a)]$. As in MBOP Argenson & Dulac-Arnold (2020), the value function is used to guide exploration and provide a terminal cost during the planning stage, which enables us to effectively extend the length of the planning horizon.

## 4.2 POLICY-GUIDED ROLLOUTS VIA THOMPSON SAMPLING

As opposed to MBOP Argenson & Dulac-Arnold (2020), L-MBOP-E samples actions from the behaviour cloned policy $f_b$, as well as from an extrinsic exploration policy $\pi_c$. The use of a secondary policy aims to boost the exploration by allowing the algorithm to sample actions that might not be sampled if we exclusively follow the behaviour policy. To determine which policy should be used during rollouts, we model the policy selection process as a *two-armed bandit problem*, and use Gaussian Thompson sampling to learn which policy performs better. In particular, we model the return of the behavior policy $x_b$ and the return of the extrinsic exploration $x_c$ policy as Gaussian distributions.

More specifically, at each iteration $t$, we are given the current state $s_t$ of the environment and initialize two sets $\mathbf{R}_b^t$ and $\mathbf{R}_c^t$ to store the associated cumulative returns from running policies $f_b$ and $\pi_c$, respectively. $N$ trajectory rollouts will be generated from the current state using the latent dynamics model $f_m$.

**Action Selection.** For the $n$-th rollout trajectory, $n = 1, ..., N$, the algorithm uses the sampled return based on parameters learned through Thompson Sampling to determine which policy should be used for the rollout:

$$\pi(s_t) = \begin{cases} f_b(s_t) & \text{if } x_b^t \geq x_c^t \\ \pi_c(s_t) & \text{otherwise} \end{cases}, \quad x_b^t \sim \mathcal{N}(\mu_b^t, \sigma_b^t), \quad x_c^t \sim \mathcal{N}(\mu_c^t, \sigma_c^t) \quad (1)$$

where $\pi_c$ is given and $f_b$ is given by:

$$f_b(s_t) = \arg\max_{a \in A_h} Q_b(z_h, a), \quad A_h = \{a_h^i\}_{i=1}^{K_Q}, \quad a_h^i \sim \mathcal{N}(\boldsymbol{\mu}(z_h), \text{diag}(\sigma_M \cdot \boldsymbol{\sigma}(z_h))^2),$$

where $\sigma_M > 0$ is a hyperparameter for scaling the standard deviation of the predicted actions. Following MBOP and MOPP, the sampled action $\pi(s_t)$ is mixed with the trajectory from the previous timestep with a mixing parameter $\beta$ to produce the action for the next step in the rollout (ref. Algorithm 1 line 11).

**Parameter Updates.** At the end of each rollout, the cumulative reward is obtained by the summation of the total return over the $H$ steps rollout and the terminal cost for the final state, $V_b(z_H)$. The total return is then added either to $\mathbf{R}_b^t$ or to $\mathbf{R}_c^t$, depending on which policy was used during the rollout. Let $n_b$ denote the number of rollouts taken from $f_b$ at iteration $t$, $N_b^t$ the total number of rollouts taken from $f_b$ up until iteration $t$, $\overline{\mathbf{R}_b^t}$ be the mean return from the set of generated rollouts, and $\mathbf{R}_b^t[i]$ the $i$-th element of the set $\mathbf{R}_b^t$. At the end of iteration $t$, we use Welford's algorithm Welford (1962) adapted for batch data to update the Gaussian distribution parameters for the policy returns:

$$\mu_b^{t+1} = \frac{N_b^t \cdot \mu_b^t + n_b \cdot \overline{\mathbf{R}_b^t}}{N_b^t + n_b}, N_b^{t+1} = N_b^t + n_b, dev_b^{t+1} = dev_b^t + n_b(\mu_b^t - \mu_b^{t+1})^2 + \sum_{i=1}^{n_b} (\mathbf{R}_b^t[i] - \mu_b^t)^2. \quad (2)$$

The standard deviation is computed as $\sigma_b^t = \sqrt{\frac{dev_b^t}{N_b^t - 1}}$. We update $\mu_c^t$, $\sigma_c^t$, and $N_c^t$ in the same way.

## 4.3 ALGORITHM DESIGN

L-MBOP-E is outlined in Algorithm 1, which follows the finite-horizon Model Predictive Control (MPC) framework. Based on the dynamics model, MPC computes a locally optimal policy by returning a sequence of actions of length $H$. At each iteration $t$, MPC executes the first planned action of the returned sequence, and then re-plans a new sequence for the newly observed state.

At each iteration $t$, L-MBOP-E performs $N$ rollouts from the current state; to determine whether the rollout is following $f_b$ or $\pi_c$, samples of policy returns are generated from the Gaussian distributions for $f_b$ and $\pi_c$, and the policy corresponding to the larger return is selected. At the end of the rollout, a terminal cost is added by using $Q_b$ or $Q_c$, respectively. After all $N$ rollouts are performed, the Gaussian distributions for the returns of $f_b$ and $\pi_c$ are updated via Welford's algorithm, and then the

---

**Algorithm 1** L-MBOP-E Algorithm

---

1: Initialize Thompson Sampling parameters: $N_b^1 = N_c^1 = 1, \mu_b^1, \mu_c^1, dev_b^1, dev_c^1 = 0$.
2: **for** $t = 1, 2 \ldots$ **do**
3:     Observe current environment state $s_t$, and encode state $s_t$ into latent state $z_t$
4:     Initialize $\mathbf{R}_b$ and $\mathbf{R}_c$ as empty sets to contain trajectory scores
5:     Set $\mathbf{A}_{N,H} = \vec{0}_{N,H}$
6:     **for** $n = 1, \ldots, N$ **do**
7:         Initialize $R_n = 0$ and sample $x_b \sim \mathcal{N}(\mu_b^t, \sigma_b^t), x_c \sim \mathcal{N}(\mu_c^t, \sigma_c^t)$
8:         Select policy $\pi_n$ based on the maximum between $x_b$ and $x_c$
9:         **for** $h = 1..H$ **do**
10:           Sample $a_h$ from policy $\pi_n$ using Eqn. equation 4.2
11:           $\mathbf{A}_{n,h} = (1-\beta)a_h + \beta \mathbf{T}_{i=\min(h,H)}, \; R_n \leftarrow R_n + f_m(z_h, \mathbf{A}_{n,h})_r, \; z_{h+1} \sim f_m(z_h, \mathbf{A}_{n,h})_s$
12:         **end for**
13:         **if** $\pi_n = f_b$ **then**
14:           Estimate $V_b(z_H)$ by sampling actions from $f_b(z_H)$ and averaging $Q_b(z_H, a)$
15:           $R_n \leftarrow R_n + V_b(z_H)$, and add $R_n$ to $\mathbf{R}_b$
16:         **else**
17:           Estimate $V_c(z_H)$ by sampling actions from $\pi_c(z_H)$ and averaging $Q_c(z_H, a)$
18:           $R_n \leftarrow R_n + V_c(z_H)$, and add $R_n$ to $\mathbf{R}_c$
19:         **end if**
20:     **end for**
21:     Update Thompson Sampling parameters for $f_b$ and $\pi_c$ using Eqn. equation 2
22:     $\mathbf{T}_h' = \frac{\sum_{n=1}^N e^{\kappa \mathbf{R}_n} \mathbf{A}_{n,h}}{\sum_{n=1}^N e^{\kappa \mathbf{R}_n}}, \forall h \in [1, H], \mathbf{R}_n = \mathbf{R}_b \cup \mathbf{R}_c$
23:     Execute action $\mathbf{T}_1'$ in real environment
24: **end for**

---

MPPI trajectory optimizer is employed to return a final trajectory from the set of $N$ rollouts, where the first action is executed for online planning in the environment.

## 5 EXPERIMENTS

To evaluate the effectiveness of L-MBOP-E, we consider the standard offline RL benchmark D4RL Fu et al. (2020) and the Deepmind Control (DMC) Tassa et al. (2018) tasks, and use state-of-the-art offline planning methods as the baselines. In what follows, we first show how L-MBOP-E performs compared to the baseline algorithms. Next, we provide a comprehensive ablation study to investigate the impact of each key design component. Finally, we evaluate the adaptability of L-MBOP-E through experiments on zero-shot adaptation for new tasks.

### 5.1 PERFORMANCE ON MUJOCO AND DEEPMIND CONTROL (DMC) TASKS

We perform experiments on three D4RL environments: halfcheetah, hopper, and walker2d, and on two DMC tasks: humanoid and quadruped. For each environment, we consider four different qualities of offline datasets (random, medium, medium-replay, and med-expert). We compare the performance of L-MBOP-E to two offline planning algorithms: MBOP Argenson & Dulac-Arnold (2020) and MOPP Zhan et al. (2021). We also consider the BC policy learnt with behavior cloning as a baseline. For convenience, the extrinsic policy is obtained as a variant by training a policy using SAC Haarnoja et al. (2018) on the same task until it performs reasonably well as the BC policy.

The results are reported in Table 1. L-MBOP-E outperforms all the baselines in almost every task. As expected, the performance gain is more significant when the quality of the offline data is lower, because the extrinsic policy is more likely to complement the BC policy for a better exploration of the state-action space.

In particular, in the case with random datasets where the BC policy is of very-low quality, BC-guided exploration is clearly not productive. As a result, both MBOP and MOPP, which only use the BC policy for guidance, degrade in performance. In contrast, L-MBOP-E can yield a substantial performance gain by training the latent dynamics model and using the Thompson Sampling exploration scheme to follow the guidance of the better policy.

| Dataset | Environment | BC | MBOP | MOPP | L-MBOP-E | MBOP-E | L-MBOP |
|---------|-------------|-----|------|------|----------|--------|--------|
| random | halfcheetah | 0.0 | 6.3 | 9.4 | **21.2** | 20.1 | 9.3 |
| random | walker2d | 0.1 | 8.1 | 6.3 | **23.7** | 19.2 | 8.0 |
| random | hopper | 0.8 | 10.8 | 13.7 | **20.2** | 18.8 | 11.3 |
| random | humanoid | 0.0 | 0.3 | - | **5.1** | 4.9 | 0.4 |
| random | quadruped | 0.1 | 4.0 | - | **11.8** | 11.1 | 3.8 |
| medium | halfcheetah | 38.9 | 44.6 | 44.7 | **56.2** | 48.6 | 55.2 |
| medium | walker2d | 60.6 | 41.0 | 80.7 | **84.7** | 82.1 | 85.6 |
| medium | hopper | 40.9 | 48.8 | 31.8 | **63.8** | 51.1 | 55.3 |
| medium | humanoid | 12.1 | 14.9 | - | **21.5** | 20.1 | 15.2 |
| medium | quadruped | 21.7 | 28.0 | - | **35.5** | 33.7 | 30.2 |
| medium-replay | halfcheetah | 27.7 | 42.3 | 43.1 | **46.8** | 43.4 | 43.9 |
| medium-replay | walker2d | 17.7 | 9.7 | 18.5 | **35.6** | 32.1 | 28.5 |
| medium-replay | hopper | 13.5 | 12.4 | 32.3 | **42.7** | 35.2 | 30.6 |
| medium-replay | humanoid | 10.8 | 15.3 | - | **19.0** | 16.7 | 15.5 |
| medium-replay | quadruped | 16.6 | 21.5 | - | **29.4** | 29.1 | 21.1 |
| med-expert | halfcheetah | 57.2 | 105.9 | **106.2** | 91.6 | 92.1 | 94.3 |
| med-expert | walker2d | 79.7 | 70.2 | 92.9 | **112.1** | 94.1 | 110.7 |
| med-expert | hopper | 50.4 | 55.1 | 95.4 | **96.7** | 96.5 | 97.1 |
| med-expert | humanoid | 14.9 | 19.6 | - | **32.8** | 30.4 | 19.9 |
| med-expert | quadruped | 87.7 | 90.1 | - | **91.2** | 90.9 | 89.9 |

Table 1: Experimental results. Scores are normalized between 0 and 100, where 100 represents the score of the expert policy. The scores for MBOP and MOPP are taken from their respective papers where possible. See appendix for full table with variance.

## 5.2 ABLATION STUDY

To clearly understand the impact of the different design choices in the proposed algorithm, we next conduct ablation studies on the hopper-medium dataset.

**Latent Dynamics Model.** We begin by examining the impact of the latent dynamics model. We first compare the performance of L-MBOP-E to that of MBOP-E which learns a standard dynamics model, to determine how the latent dynamics model helps to improve planning. As shown in Table 1, L-MBOP-E clearly outperforms MBOP-E by leveraging the latent dynamics model. To further justify this, we also compare the performance between MBOP and L-MBOP with the latent model in Table 1, where substantial performance gains can be achieved in L-MBOP by using the latent model to replace the standard dynamics model.

To understand how the dimension of the latent space affects the performance, we run the experiments with different values of the size of the latent dimension, with dataset size set to 50,000. The results are reported in Figure 2a, which show that the algorithm performs well on most latent dimension sizes where the latent model is sufficient to capture the main characterizations of the dynamic model.

Another benefit of leveraging the latent model is to improve the data efficiency. To justify this, we next conduct experiments under different sizes of the dataset, ranging from 20,000 to 1,000,000 samples, fixed with latent dimension size of 9. We observe that even with a smaller training dataset, L-MBOP-E can outperform MBOP, as demonstrated by MBOP achieving a score of 1578 on 1,000,000 samples. By learning a latent space that captures the features for the environment dynamics, L-MBOP-E can attain higher data efficiency than attempting to learn the dynamics in the original space.

**Benefits of Using the Extrinsic Policy.** First, to understand the benefit of the extrinsic policy, we compare the performance between L-MBOP and L-MBOP-E. As seen in Table 1, when Thompson Sampling is used to explore with an additional extrinsic policy, L-MBOP-E can leverage the extrinsic policy to provide additional guidance for planning and improves upon L-MBOP.

Next, we investigate how the quality of the extrinsic policy affects the performance of the proposed algorithm L-MBOP-E. To this end, we run the experiments under different qualities of the extrinsic policy, i.e., low, medium, medium-expert and expert, and compare to just using L-MBOP; the results are shown in Figure 3b. Our findings indicate that increasing the extrinsic policy quality from low to

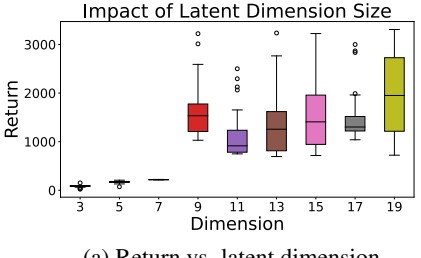 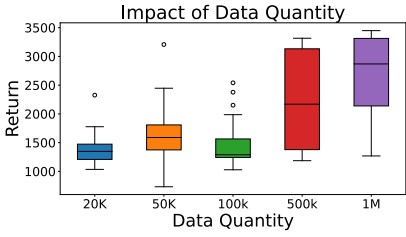

(a) Return vs. latent dimension                    (b) Return vs. dataset sizes

Figure 2: Performance of L-MBOP-E trained with varying amounts of data and latent dimension sizes. In Figure 2a, the size of the latent dimension is varied between 3 and 19, in the case where the dataset size is 50,000. In Figure 2b, the dataset is set to sizes between 20,000 and 1,000,000, and the latent dimension is fixed to 9.

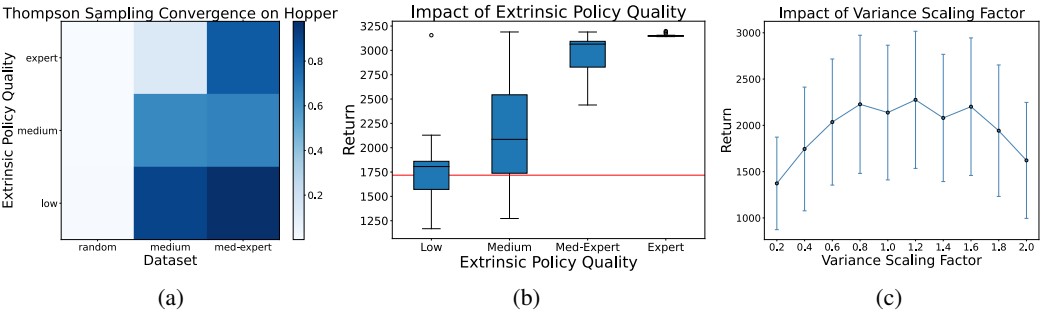

(a)                              (b)                              (c)

Figure 3: (a) Converged $p_t$ value from Thompson Sampling vs. various qualities of datasets and extrinsic policies. The darker the shade, the higher the converged $p_t$ value (higher probability of sampling from BC policy). (b) Performance of L-MBOP-E with varying qualities of the extrinsic policy. The red line represents the score of L-MBOP as a baseline. (c) Sensitivity to the variance scaling factor $\sigma_M$.

expert consistently improve the quality of the planning. By using Thompson Sampling, even in the case where we use a low-quality extrinsic policy, L-MBOP-E can selectively sample actions from it and still improve upon the overall performance of L-MBOP. As the quality of the extrinsic policy increases, $p_t$ will get closer to 0 and the algorithm learns to follow the extrinsic policy.

**Impact of Thompson Sampling.** We conduct experiments under different qualities of the offline dataset and extrinsic policy in order to determine whether Thompson Sampling can correctly identify which policy is stronger. Intuitively, Thompson Sampling should converge to a low $p_t$ value when the extrinsic policy is better than the BC policy and a high $p_t$ value when the BC policy is better. As shown in Figure 3a, in the random dataset where the performance of the BC policy is of low quality, the value of $p_t$ converges to near 0.0 so that the exploration would tend to follow the guidance of the extrinsic policy. In the other extreme case where the BC policy is of high quality and the extrinsic policy is of low quality, $p_t$ will converge to a value close to 1.0.

Next, we investigate the impact of the variance scaling factor $\sigma_M$ on the learning performance. Specifically, for the Hopper-medium task, we test different variance scale values ranging from 0.2 to 2.0. As shown in Figure 3c, as we increase $\sigma_M$, the performance initially increases because the algorithm will sample more diverse actions for better exploration. The performance decreases if $\sigma_M$ is too large, because of the increased distributional shift between the sampled actions and the BC policy. However, the performance of our algorithm is robust to the selection of $\sigma_M$.

The Thompson Sampling algorithm allows L-MBOP-E to sample actions from both the BC policy and extrinsic policy, and follow the better policy for different states. To verify this, we perform Principal Component Analysis on a sample trajectory from the Hopper environment. In order to distinguish which states the extrinsic policy performs better, we use the $N$ trajectory rollouts generated by L-MBOP-E during planning. We compute the average return of the trajectories generated by either policy, and color red the states where the extrinsic policy has higher average return. The visualizations are shown in Figure 4a and Figure 4b. As we can see, the latent state space has a

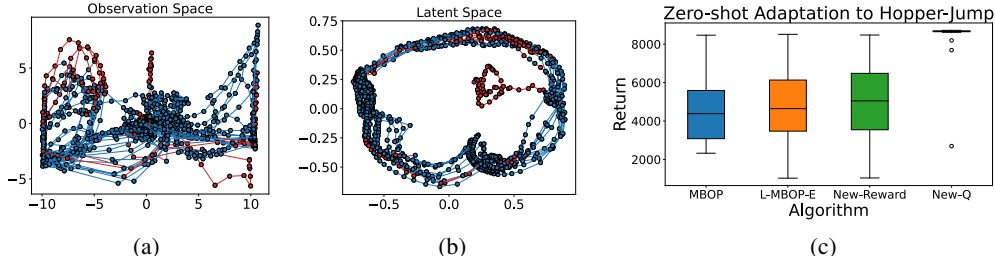

Figure 4: (a-b) Visualization of states visited in the observation and latent spaces. States where the extrinsic policy outperforms the BC policy are colored red. (c) Zero-shot adaptation experiments for Hopper-Jump. It can be seen that L-MBOP-E clearly outperforms MBOP by using the new reward and the improvement is more significant when a new Q function can be trained with the new reward.

very clear structure which makes the characterization of the dynamics easier and more accurate. Moreover, it is clear that the planning deviates from the BC policy and follows the guidance of the extrinsic policy in the red states, leading to the improved overall performance.

### 5.3 ZERO-SHOT TASK ADAPTATION

One of the main advantages to using the model-based planning framework is the ability to adapt to new reward signals without having to re-train a policy. We can use a new reward signal by simply replacing the predicted reward received during the synthetic rollouts with the new reward function. This will allow L-MBOP-E to optimize the trajectories according to the new reward function, and enable zero-shot task adaptation.

To verify this, we consider a new task, Hopper-Jump, which uses the original Hopper environment but encourages the agent to jump by rewarding the agent for its z-position. The new reward function is defined as follows:

$$r_{new} = \alpha_r \cdot r_{original} + (1 - \alpha_r) \cdot 10 \cdot z$$

where $r_{original}$ is the reward function from Hopper, $z$ is the z-position of the Hopper, and $\alpha_r \in [0, 1]$ is a mixing parameter, which we set to $\alpha_r = 0.5$.

Because the Q function was learnt based on the reward function from the offline dataset $\mathcal{D}$, using it as the terminal cost could degrade the performance if the new reward for the modified environment differs greatly from the reward in the dataset. A new Q function can be trained from the offline dataset by using the new reward function to replace the original reward. This allows L-MBOP-E to use a more accurate terminal cost for better adaptation performance.

We compare the performance of MBOP and L-MBOP-E, along with two variants of L-MBOP-E: L-MBOP-E with the new reward function (New-Reward), and L-MBOP-E with both the new reward function and the Q function re-trained on the new reward (New-Q). Results are plotted in Figure 4c. As shown, L-MBOP-E improves upon the performance of MBOP. Using the new reward function allows L-MBOP-E to optimize trajectories for the new task to improve the performance. If a new Q function is further trained, long-term consequences can be considered and the planning quality is greatly improved. All these results clearly demonstrate the superior zero-shot adaptation capability of L-MBOP-E, which is built on the latent model learning and better exploration with Thompson Sampling and the extrinsic policy.

## 6 CONCLUSION

We develop Latent-Model Based Offline Planning with Extrinsic Policy-Guided Exploration (L-MBOP-E), which is built on two key ideas: 1) low-dimensional latent model learning to reduce the effects of compounding errors when learning a dynamics model with limited offline data, and 2) a Thompson Sampling based exploration strategy with an extrinsic policy to guide planning beyond the behavior policy and hence get the best out of these two policies. Experimental results demonstrate that L-MBOP-E significantly outperforms the state-of-the-art algorithms on the D4RL and DMC tasks, and performs especially well when given access to an extrinsic policy which complements the BC policy.

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

## A    APPENDIX

## B    EXPERIMENT SETTINGS

### B.1    DATASETS

We evaluate L-MBOP-E on the popular D4RL and DMC tasks. In D4RL we use the tasks halfcheetah, walker2d, and hopper. For DMC, we experiment with the more difficult humanoid-walk and quadruped-walk tasks. On each task we train L-MBOP-E on four different datasets: random, medium, medium-replay, and med-expert. The datasets are generated from following behavior policies of various qualities:

- **random** - a collection of 1M steps is obtained by rolling out a randomly-initialized policy.
- **medium** - a policy is trained via SAC to one-third the performance of the expert policy, then is rolled out for 1M steps.
- **medium-replay** - the dataset is the replay buffer from the medium quality policy.
- **med-expert** - this dataset is the union of the medium dataset and an additional 1M samples generated by following the expert policy.

For the DMC tasks, we create our own datasets following the above method.

### B.2    MODEL CONFIGURATIONS

The dynamics model, behavior cloned policy, and Q function are all approximated as feed-forward neural networks. We use the following configurations for all experiments.

#### B.2.1    LATENT DYNAMICS MODEL

The latent dynamics model $f_m$ consists of a state encoder, state decoder, and transition model. All three networks are configured as follows:

- Number of Hidden Layers: 2
- Size of Hidden Layers: 500
- Activation: ReLu
- Learning Rate: 0.0001
- Batch Size: 128
- Epochs: 40
- Optimizer: Adam

We fix the size of the latent dimension depending on the task:

- HalfCheetah: 15
- Walker2d: 15
- Hopper: 9

#### B.2.2    BEHAVIOR POLICY AND Q FUNCTION

The behavior policy $f_b$ and Q function $Q_b$ share the same configuration:

- Number of Hidden Layers: 2
- Size of Hidden Layers: 500
- Activation: ReLu
- Learning Rate: 0.001
- Batch Size: 512
- Epochs: 40
- Optimizer: Adam

| HalfCheetah | | | | | |
| --- | --- | --- | --- | --- | --- |
| Dataset | $H$ | $\kappa$ | $\beta$ | $\sigma_M$ | $N$ |
| random | 4 | 3 | 0 | 0.75 | 100 |
| medium | 2 | 3 | 0 | 1.0 | 100 |
| medium-replay | 4 | 3 | 0 | 1.0 | 100 |
| med-expert | 2 | 1 | 0 | 0.7 | 100 |

| Walker2d | | | | | |
| --- | --- | --- | --- | --- | --- |
| Dataset | $H$ | $\kappa$ | $\beta$ | $\sigma_M$ | $N$ |
| random | 8 | 0.3 | 0 | 1.0 | 1000 |
| medium | 2 | 0.1 | 0 | 1.0 | 1000 |
| medium-replay | 8 | 3 | 0 | 0.3 | 1000 |
| med-expert | 2 | 1 | 0 | 1.0 | 1000 |

| Hopper | | | | | |
| --- | --- | --- | --- | --- | --- |
| Dataset | $H$ | $\kappa$ | $\beta$ | $\sigma_M$ | $N$ |
| random | 4 | 10 | 0 | 0.5 | 100 |
| medium | 4 | 0.3 | 0 | 1.2 | 100 |
| medium-replay | 4 | 0.3 | 0 | 1.5 | 100 |
| med-expert | 10 | 3 | 0 | 0.3 | 100 |

| Humanoid | | | | | |
| --- | --- | --- | --- | --- | --- |
| Dataset | $H$ | $\kappa$ | $\beta$ | $\sigma_M$ | $N$ |
| random | 4 | 4 | 0 | 0.2 | 100 |
| medium | 4 | 1 | 0 | 0.2 | 100 |
| medium-replay | 4 | 1 | 0 | 0.2 | 100 |
| med-expert | 4 | 2 | 0 | 0.2 | 100 |

| Quadruped | | | | | |
| --- | --- | --- | --- | --- | --- |
| Dataset | $H$ | $\kappa$ | $\beta$ | $\sigma_M$ | $N$ |
| random | 4 | 4 | 0 | 0.3 | 100 |
| medium | 4 | 2 | 0 | 0.3 | 100 |
| medium-replay | 4 | 2 | 0 | 0.3 | 100 |
| med-expert | 4 | 2 | 0 | 0.3 | 100 |

Table 2: Hyperparameter configurations for experiments

### B.3 PLANNING HYPERPARAMETERS

We present the selected hyperparameters used in each task for L-MBOP-E in Table 2. We select hyperparameters similar to the ones used by MBOP and MOPP to make the results more comparable. We also use these same parameters during the ablation studies, expect for the parameter being varied. Meanwhile, we set $K_Q = 10$ in Eqn. equation 4.2 for all experiments.

| Dataset | Environment | BC | MBOP | MOPP | L-MBOP-E |
|---|---|---|---|---|---|
| random | halfcheetah | 0.0±0.0 | 6.3±4.0 | 9.4±2.6 | **21.2±6.4** |
| random | walker2d | 0.1±0.1 | 8.1±5.5 | 6.3±0.1 | **23.7±2.5** |
| random | hopper | 0.8±0.6 | 10.8±0.3 | 13.7±2.5 | **20.2±2.4** |
| medium | halfcheetah | 38.9±3.7 | 44.6±0.8 | 44.7±2.6 | **56.2±0.6** |
| medium | walker2d | 60.6±22.6 | 41.0±29.4 | 80.7±1.0 | **84.7±2.9** |
| medium | hopper | 40.9±6.9 | 48.8±26.8 | 31.8±1.3 | **63.8±19.2** |
| medium-replay | halfcheetah | 27.7±9.0 | 42.3±0.9 | 43.1±4.3 | **46.8±12.7** |
| medium-replay | walker2d | 17.7±18.4 | 9.7±5.3 | 18.5±8.4 | **35.6±14.6** |
| medium-replay | hopper | 13.5±12.6 | 12.4±5.8 | 32.3±5.9 | **42.7±12.9** |
| med-expert | halfcheetah | 57.2±22.3 | 105.9±17.8 | **106.2±5.1** | 91.6±10.2 |
| med-expert | walker2d | 79.7±26.3 | 70.2±36.2 | 92.9±14.1 | **112.1±0.9** |
| med-expert | hopper | 50.4±21.3 | 55.1±44.3 | 95.4±28.0 | **96.7±18.7** |

Table 3: Experimental results with variances.

## B.4 EXPERIMENTAL DETAILS

### B.4.1 BOOTSTRAPPING FOR Q LEARNING

We train the Q function for the behavior policy $\pi_b$ using the dataset $\mathcal{D}$ via Fitted Q Evaluation (FQE). The dataset consists of rollouts generated by $\pi_b$ in the environment; each episode terminates at an unhealthy state or continues until timeout is reached.

When applying FQE to learn the Q function for $\pi_b$, we compute the targets for the next iteration of FQE by bootstrapping the current Q-values. This can cause issues on the boundary between episodes, where we may not be able to bootstrap the future Q-value when computing the Q-value for the final state of the episode. We handle the case when an episode ends due to termination or timeout separately:

- If an episode ends due to reaching an unhealthy state, then we do not bootstrap the Q-values for the next state-action pair, and instead set the target to be the immediate reward.

- If an episode ends due to timeout, then we are not able to bootstrap because we do not have access to which state-action pair follows the final state-action pair in the episode. In this case, we treat the second-to-last timestep in the episode as the final timestep, and use the actual final timestep for bootstrapping.

Training the Q function in this manner allows for the network to learn a more accurate approximation that takes into account whether the episode ended due to termination or timeout.

## C EXPERIMENTAL RESULTS

### C.1 EXPERIMENTAL RESULTS WITH VARIANCE

We report the experimental results for L-MBOP-E with standard deviations in Table 3. Results for the ablation studies of L-MBOP-E are reported in Table 4.

### C.2 ADDITIONAL EXPERIMENTAL RESULTS

We carry out additional experiments which could not be included in the main paper due to space limitations.

**Effectiveness of Decreasing the Latent Dimension.** We first test the effectiveness of decreasing the latent dimension as a means of regularization. Our intuition is as follows: Using a lower-dimensional latent state space already implicitly applies a form of regularization to the model, and additional regularization may or may not improve the performance. Therefore, we should expect that using $L_2$ regularization may cause a drop in performance, as the learned model may underfit the dataset.

| Dataset | Environment | L-MBOP-E | MBOP-E | L-MBOP |
|---------|-------------|----------|--------|--------|
| random | halfcheetah | **21.2**±6.4 | 20.1±6.1 | 9.3±1.1 |
| random | walker2d | **23.7**±2.5 | 19.2±3.1 | 8.0±0.4 |
| random | hopper | **20.2**±2.4 | 18.8±2.2 | 11.3±0.4 |
| medium | halfcheetah | **56.2**±0.6 | 48.6±0.7 | 55.2±0.2 |
| medium | walker2d | **84.7**±2.9 | 82.1±2.7 | 85.6±13.7 |
| medium | hopper | **63.8**±19.2 | 51.1±18.6 | 55.3±27.7 |
| medium-replay | halfcheetah | **46.8**±12.7 | 43.4±12.2 | 43.9±0.3 |
| medium-replay | walker2d | **35.6**±14.6 | 32.1±15.2 | 28.5±9.1 |
| medium-replay | hopper | **42.7**±12.9 | 35.2±11.5 | 30.6±2.8 |
| med-expert | halfcheetah | 91.6±10.2 | 92.1±11.4 | 94.3±10.6 |
| med-expert | walker2d | **112.1**±0.9 | 94.1±1.1 | 110.7±0.3 |
| med-expert | hopper | **96.7**±18.7 | 96.5±17.7 | 97.1±19.2 |

Table 4: Ablation study results with variances.

| Dataset | Environment | BC | MBOP | MOPP | L-MBOP-E |
|---------|-------------|-----|------|------|----------|
| random | halfcheetah | 0.0±0.0 | 6.3±4.0 | 9.4±2.6 | **21.3**±6.6 |
| random | walker2d | 0.1±0.1 | 8.1±5.5 | 6.3±0.1 | **23.6**±2.4 |
| random | hopper | 0.8±0.6 | 10.8±0.3 | 13.7±2.5 | **20.2**±2.1 |
| medium | halfcheetah | 38.9±3.7 | 44.6±0.8 | 44.7±2.6 | **56.6**±0.6 |
| medium | walker2d | 60.6±22.6 | 41.0±29.4 | 80.7±1.0 | **83.3**±3.8 |
| medium | hopper | 40.9±6.9 | 48.8±26.8 | 31.8±1.3 | **63.7**±21.3 |
| medium-replay | halfcheetah | 27.7±9.0 | 42.3±0.9 | 43.1±4.3 | **50.7**±1.1 |
| medium-replay | walker2d | 17.7±18.4 | 9.7±5.3 | 18.5±8.4 | **36.7**±10.6 |
| medium-replay | hopper | 13.5±12.6 | 12.4±5.8 | 32.3±5.9 | **42.2**±13.3 |
| med-expert | halfcheetah | 57.2±22.3 | 105.9±17.8 | **106.2**±5.1 | 101.8±4.4 |
| med-expert | walker2d | 79.7±26.3 | 70.2±36.2 | 92.9±14.1 | **112.9**±0.7 |
| med-expert | hopper | 50.4±21.3 | 55.1±44.3 | 95.4±28.0 | **95.9**±16.9 |

Table 5: Experimental results without $L_2$ regularization.

To this end, we train the latent dynamics model of L-MBOP-E without $L_2$ regularization, and report the performance results in Table 5. We observe that in some cases, Without $L_2$ regularization in fact results in slightly improved performance on certain tasks. On most tasks, the performance does not improve by much. However, on the halfcheetah environment there is a noticeable improvement in the performance, especially in the medium-replay and med-expert datasets.

**Visualization of the Latent State Space.** The visualization of the structure of the learnt latent space for the halfcheetah, hopper, and walker2d environments are displayed in Figure 6. Contrasting the observation space with the latent space, we can see a clear structure emerges when we embed the observations into a latent space. We can see that the latent space captures features which lead to a more structured and predictable approximation of the dynamics.

**Thompson Sampling.** We conduct experiments to determine whether Thompson Sampling can identify which policy between the BC policy $f_b$ and the extrinsic policy $\pi_c$ performs better. Experiments are conducted for the halfcheetah, walker2d, and hopper environments where we vary the quality of the dataset and the quality of the extrinsic policy. We report the converged $p_t$ value in every case in Figure 5. As can be seen, when the BC policy is stronger than the extrinsic policy, the converged $p_t$ value will be close to 1, in order to promote sampling from the BC policy. In the other case where the extrinsic policy is better performing than the BC policy, $p_t$ will converge to a value near 0. Finally, when the policies are of similar quality, $p_t$ converges to a value near 0.50 to allow sampling from both policies.

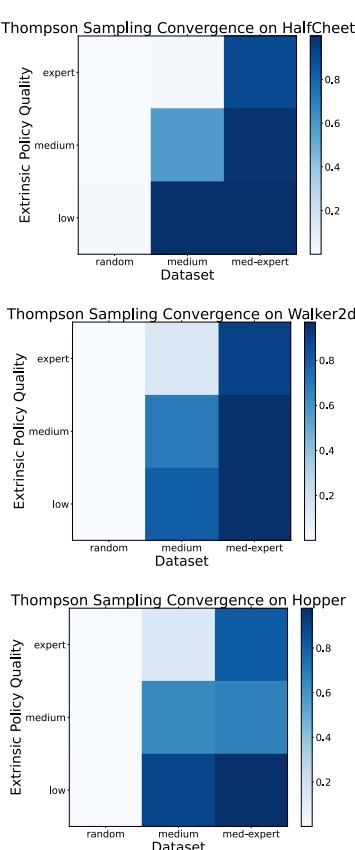

Figure 5: Converged $p_t$ value from Thompson Sampling vs. various qualities of datasets and extrinsic policies. The darker the shade, the larger the converged $p_t$ value.

State visualizations for HalfCheetah task

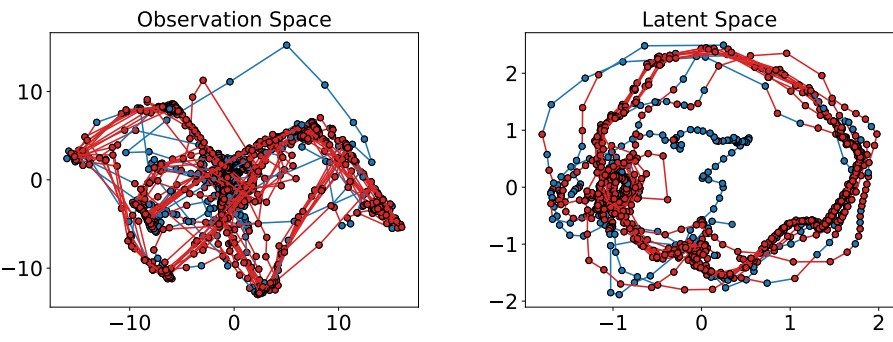

State visualizations for Walker2d task

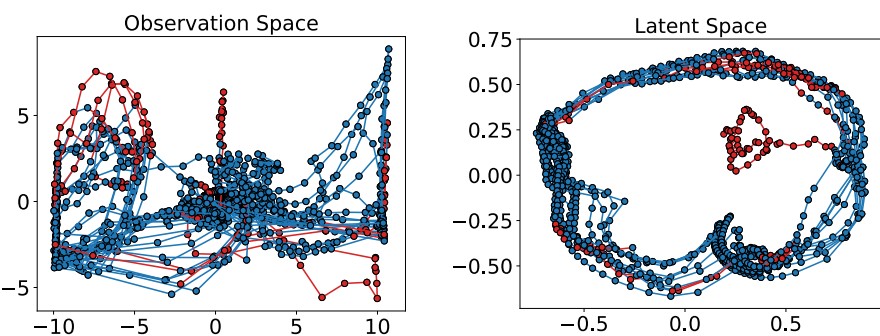

State visualizations for Hopper task

Figure 6: Visualization of visited states in a sample trajectory for the halfcheetah, walker2d, and hopper tasks. Results are shown for both the original observation space and the learnt latent space.

## D    EXECUTION SPEED

Execution speeds for L-MBOP-E are reported in Table 6, with simulation time included. We test the execution speed for the walker2d task where we fix $N = 1000$, and the hopper task, where we set $N = 100$. The planning horizon is varied between $H = 2$ and $H = 16$. We run all experiments on an Ampere A100 GPU, using a single core of an AMD EPYC 7513 CPU. We note that L-MBOP-E can achieve high control frequencies and is suitable for real-time use. If high-frequency control is required, the planning horizon $H$ or the number of rollouts $N$ can be reduced to increase the execution speed.

| $H$ | Frequency (Hz) | |
| --- | --- | --- |
| | Walker2d | Hopper |
| 2 | 133 | 134 |
| 4 | 75 | 77 |
| 8 | 27 | 46 |
| 16 | 18 | 24 |

Table 6: Control frequency of L-MBOP-E on the walker2d and hopper tasks with varying planning horizons.

