# OpenReview forum: "L-MBOP-E: Latent-Model Based Offline Planning with Extrinsic Policy Guided Exploration"
_ICLR.cc/2024/Conference — Submitted to ICLR 2024_

### Official Review · Reviewer_4zzv · 2023-10-29

**Soundness:** 2 fair
**Presentation:** 2 fair
**Contribution:** 1 poor
**Rating:** 3
**Confidence:** 4

**Summary:**

This paper introduces a learning technique for model-based offline planning that employs a latent state representation and an extrinsic policy to supplement the behavior policy during planning. The proposed algorithm, L-MBOP-E, is evaluated in comparison to two other model-based offline RL methods, MBOP and MOPP, on  several tasks from the D4RL and DMC datasets. Experimental results demonstrate that L-MBOP-E outperforms other model-based offline planning methods.

**Strengths:**

1. This paper is well written and easy to follow.
2. The idea of using Thompson Sampling in the context of model-based planning seems novel and interesting.

**Weaknesses:**

1. Although the authors claim in the introduction that "we advocate a low-dimensional latent representation for the state space, which can yield a higher accuracy dynamics model and in turn improve prediction quality and hence reduce the likelihood of compounding errors," there are no experimental results or relevant papers to support this view.

2. In visual model-based RL, the latent world model has already been widely used. The latent model proposed by the author, apart from replacing Dreamer's RSSM with MLP, has no fundamental difference from Dreamer [1]. Moreover, Dreamer also supports experiments with state input. Thus, I believe that merely claiming the use of a latent model in offline RL lacks novelty.

3. The biggest issue with this paper is that it does not explain how the extrinsic policy used for exploration is trained or what dataset is used. If it's merely learning a policy online in the same environment, this undoubtedly constitutes cheating in the offline setting. If a policy is learned from a different environment, how can one avoid the model error caused by the unseen actions selected by this policy during offline planning?

4. The paper doesn't compare with model-based offline RL methods which also aim to solve inaccurate models and overly conservative, such as MOPO [2], MORel [3], COMBO [4], and CBOP [5].


Reference:

[1]. DREAM TO CONTROL: LEARNING BEHAVIORS BY LATENT IMAGINATION. Hafner et. al., ICLR 2020.

[2]. MOPO: Model-based Offline Policy Optimization. You et. al., NeurIPS 2020.

[3]. MOReL: Model-Based Offline Reinforcement Learning. Kidambi et. al., NeurIPS 2020.

[4]. COMBO: Conservative Offline Model-Based Policy Optimization. You et. al., NeurIPS 2021.

[5]. CONSERVATIVE BAYESIAN MODEL-BASED VALUE EXPANSION FOR OFFLINE POLICY OPTIMIZATION. Jeong et. al., ICLR 2023.

**Questions:**

1. In the section "Behavior Policy and Value Function Learning in the Latent Space," the authors did not use common methods like CQL [1] to train the critic. So, how does this critic avoid overestimation or underestimation for unseen state-action pairs?

2. I am curious about the comparison with uncertain regularized planning methods. The paper does not mention this, but using a model/critic ensemble to guide exploration or to avoid high error areas during the rollout/planning process is already a standard practice [2] [3].

3. The paper claims that the proposed latent model can be used for zero-shot adaptation, but PlaNet's world model can as well. Therefore, I strongly recommend that the authors add a comparison with PlaNet [4].

Reference:

[1]. Conservative Q-Learning for Offline Reinforcement Learning. Kumar et. al., NeurIPS 2020.

[2]. Learning to Plan Optimistically: Uncertainty-Guided Deep Exploration via Latent Model Ensembles. Seyde et. al., CoRL 2021.

[3]. Finetuning Offline World Models in the Real World. Feng et. al., CoRL 2023.

[4]. Learning Latent Dynamics for Planning from Pixels. Hafner et. al., ICML 2019.

---

### Official Review · Reviewer_yied · 2023-10-31

**Soundness:** 1 poor
**Presentation:** 3 good
**Contribution:** 1 poor
**Rating:** 3
**Confidence:** 4

**Summary:**

In this paper, it presents L-MBOP-E, a model-based offline reinforcement learning algorithm that constructs a latent dynamics model from an offline dataset. It operates under the assumption that a viable extrinsic policy is accessible to guide policy optimization. The key idea is the utilization of Thompson sampling as a method to decide the usage between an extrinsic policy and a behavior-cloned policy for trajectory optimization. In this way, it imposes constraints on the selected actions while avoiding excessive pessimism. Experiments on D4RL benchmarks demonstrate the effectiveness of the proposed approach

**Strengths:**

The paper is well-written so that it is easy to follow the proposed method.

**Weaknesses:**

The method relies on a strong assumption that there exists a non-trivial extrinsic policy in addition to the behavior policy which collects the offline dataset so that it could guide the agent’s learning process to avoid over pessimism. However, in general such an extrinsic policy could be hard to obtain in many realistic settings.

The usage of latent dynamics model is over-claimed as a novel contribution. The application of latent dynamics models shows significant gain in the experiments. However, given the related model-based RL works such as PlaNet, Dreamer (v1-v3), TDMPC, the learning objective of the latent dynamics model is not novel at all.

A primary advantage of latent dynamics models is their ability to facilitate learning from high-dimensional pixel observations. The absence of visual offline reinforcement learning experiments in the current study is a notable omission, representing a significant constraint of the experimental framework. However, no visual offline RL experiments are conducted in the paper. This is a major limitation of the experiments. Furthermore, many other recent offline RL algorithms are not compared. (See Questions for details.)

**Questions:**

1. I don’t get this sentence, which describes the difference between the proposed method and the prior latent dynamics model based approach “Different from the above approaches, L-MBOP-E employs a state decoder, …”.  Why is the latent dynamics model objective claimed to be novel? Dreamer-v2/PlaNet also employs a state decoder, and it proposes a recurrent RSSM based dynamics model to deal with partial observability in the latent space. Neither the learning objective nor the architecture in this paper is novel.

2. Why not jointly learn policy & value functions and latent dynamics models instead of relying on state decoder/reconstruction? The usage of a state decoder is in fact a big limitation here because this objective actually tries to model every dimension of the input without taking into account which dimensions are more important and which dimensions are not. This is exactly the value-equivalent principle in model-based RL [1,2], and part of the reason why TDMPC also learns a Q function on top of the embedding spaces, which can serve to shape the representation. So the latent representations learned in this approach could be highly suboptimal.
3. Some recent offline RL algorithms such as [5][6][7][8] are ignored in the comparison.
4. (**More important**) Empirical evaluations of offline visual RL should be conducted. [9][10][11] are all recent works on visual offline RL that should be compared and discussed in the paper.

References:
- [1] Grimm et al., The Value Equivalence Principle for Model-Based Reinforcement Learning, NeurIPS 2020.
- [2] Amir-Massoud Farahmand et al., Value-Aware Loss Function for Model-based Reinforcement Learning Amir-Massoud, AISTATS 2018
- [5] Kostrikov et al. Offline Reinforcement Learning with Implicit Q-Learning, ICLR 2022
- [6] Fujimoto et al. A Minimalist Approach to Offline Reinforcement Learning, NeurIPS 2021
- [7] Ching-An et al. Adversarially Trained Actor Critic for Offline Reinforcement Learning, ICML 2022
- [8] Kidambi et al. MOReL: Model-Based Offline Reinforcement Learning, NeurIPS 2020
- [9] Lu et al. Challenges and Opportunities in Offline Reinforcement Learning from Visual Observations
- [10] Zheng et al. TACO: Temporal Latent Action-Driven Contrastive Loss for Visual Reinforcement Learning, NeurIPS 2023
- [11] Islam et al. Principled Offline RL in the Presence of Rich Exogenous Information. ICML 2023

---

### Official Review · Reviewer_sSEa · 2023-10-31

**Soundness:** 2 fair
**Presentation:** 3 good
**Contribution:** 2 fair
**Rating:** 5
**Confidence:** 4

**Summary:**

This paper proposes a new model-based offline planning algorithm called L-MBOP-E. The key ideas are:

Use a low-dimensional latent state representation when learning the dynamics model from offline data. This helps reduce compounding errors during planning.

Use an extrinsic policy in addition to the behavior cloned (BC) policy to guide exploration when planning trajectories. A Thompson sampling strategy determines which policy to follow.

The method learns a latent dynamics model, BC policy, and Q-function from offline data. During planning, rollouts selectively follow the BC or extrinsic policies based on Thompson sampling. The trajectory is optimized using MPPI.

**Strengths:**

Achieves state-of-the-art results on D4RL and DMC benchmarks, significantly outperforming prior offline planning methods like MBOP and MOPP.

Reduced model uncertainty and improved data efficiency by using a latent dynamics model.

Allows more flexible exploration by using an extrinsic policy in addition to the BC policy. Thompson sampling enables selectively following the better policy.

Demonstrates improved zero-shot task adaptation by using the new reward with the latent dynamics model.

**Weaknesses:**

The extrinsic policy is assumed to be given, rather than learned.

The latent dynamics model relies on a fixed size latent state chosen a priori.

No ablation on the number of rollouts N or planning horizon H was provided.

The computational complexity and wall-clock runtime of the method should be analyzed.

**Questions:**

The zero-shot transfer results require more analysis and discussion. Please provide more details on how the reward functions were modified and tuned for the new tasks. This will help assess the feasibility of the approach.

Please analyze the runtime performance of L-MBOP-E, including how it scales with key hyperparameters like the number of rollouts and planning horizon. This practical understanding of complexity is currently missing.

We suggest examining if an adaptive approach could be beneficial for determining the latent state dimensionality. This could improve the robustness of the method to this key hyperparameter.

Additional implementation details are needed on how autoencoder training was incorporated with the latent dynamics model - was it pretrained separately, or trained jointly end-to-end?

How was the extrinsic policy initialized - from the BC policy or trained from scratch?

What criteria was used to determine the latent dimensions for each environment? Was any sensitivity analysis performed?

What is the source of the variance in results across random seeds - is it due to model uncertainty or policy optimization?

Were any alternatives explored for incorporating the extrinsic policy, other than Thompson sampling?

Were planning hyperparameters like horizon and rollouts tuned per environment or fixed?

**Details Of Ethics Concerns:**

No concerns

---

### Official Review · Reviewer_763y · 2023-11-01

**Soundness:** 1 poor
**Presentation:** 3 good
**Contribution:** 2 fair
**Rating:** 3
**Confidence:** 3

**Summary:**

The paper builds on the framework of model-based offline planning (MBOP) and introduces two modifications:
* The dynamics model utilizes a latent space for prediction, aiming to mitigate the difficulty of accurately predicting dynamics in high dimensions.
* The planner incorporates an additional “extrinsic” policy which may sample actions different from the behavior policy, and Thompson sampling is employed to choose which policy to use for action execution.

The algorithm is compared to previous MBOP-style algorithms on the D4RL benchmark and DeepMind Control tasks, and the contributions of different components of the algorithm are studied via ablation.

**Strengths:**

* Each of the proposed modifications appears to benefit the algorithm compared to previous MBOP-style algorithms, and they can be combined for further performance gains.
* The incorporation of an extrinsic policy is original and potentially allows for flexible reuse of previously trained policies, extending the possibilities for MBOP-style algorithms. This could be significant for future work.
* The writing of the paper is clear.

**Weaknesses:**

Latent dynamics models have been used in prior works, so the main contribution of this paper seems to be the incorporation of an extrinsic policy. Unfortunately, I find that the paper as written has significant issues in this regard:
* While the motivation stated in the paper is that “the extrinsic policy can be a meta-learned policy or a policy learned from another similar RL task”, the experiments do not explore either of these possibilities at all. Instead, “For convenience, the extrinsic policy is obtained as a variant by training a policy using SAC on the same task until it performs reasonably well as the BC policy.” It is not clear why in practice one would have such a policy, nor how it would be similar to a meta-learned policy or a policy from another task, so the experiments do not reflect the intended use case of the proposed algorithm.
* The use of Thompson sampling is not properly ablated. The authors show that Thompson sampling tends to prefer the stronger algorithm (which is not surprising), but the impact of this sampling bias on overall algorithm performance is not shown. A simple baseline would be to always sample $N/2$ rollouts from each policy, then apply the MPPI trajectory optimizer as before.

**Questions:**

To me, the most obvious candidate for an alternative policy to the BC policy would be one trained using some offline RL algorithm. This is arguably not “extrinsic” since it depends on the same dataset, but at least it may provide different behavior than the BC policy, and it is always available. Did you experiment with anything like this?

---

### Meta-Review · Area_Chair_kvF6 · 2023-12-05

**Metareview:**

This work introduces innovations on top of the MBOP algorithm, namely using a latent dynamics model and then combining an extrinsic policy with planning. These may well be important contributions but the paper does not make a strong case for their necessity - most notably there are lacking baselines when there have been several model-based offline RL algorithms proposed since MBOP, particularly in visual model-based offline RL (e.g. using VD4RL). In addition, the reviewers noted that the introduction of a latent dynamics model was not especially novel since it has been used for many years in works such as PlaNet and Dreamer, while there were also questions about the availability of the extrinsic policy. Since this is exclusively a methods paper it would be important to have a more clear cut experiment to justify these additional modifications.

**Justification For Why Not Higher Score:**

This is a methods paper adding components to a baseline (MBOP, ICLR 2021) and providing some relatively small gains on the D4RL benchmark from 2020. The work seems a little late for this type of contribution, which may have been interesting shortly after MBOP, but three years later there have been many new methods in this space and the field has moved on (e.g. to visual observations).

**Justification For Why Not Lower Score:**

N/A

---

### Decision · Program_Chairs · 2024-01-16

Reject